# Rapid Detection of Different Types of Soil Nitrogen Using Near-Infrared Hyperspectral Imaging

**DOI:** 10.3390/molecules27062017

**Published:** 2022-03-21

**Authors:** Zhuoyi Chen, Shijie Ren, Ruimiao Qin, Pengcheng Nie

**Affiliations:** 1College of Biosystems Engineering and Food Science, Zhejiang University, Hangzhou 310058, China; zhuoyichen@zju.edu.cn (Z.C.); 22014174@zju.edu.cn (S.R.); rmqin@zju.edu.cn (R.Q.); 2Key Laboratory of Sensors Sensing, Ministry of Agriculture, Zhejiang University, Hangzhou 310058, China; 3State Key Laboratory of Modern Optical Instrumentation, Zhejiang University, Hangzhou 310058, China

**Keywords:** soil, ammonium nitrogen, nitrate nitrogen, urea nitrogen, near-infrared hyperspectral image

## Abstract

Rapid and accurate determination of soil nitrogen supply capacity by detecting nitrogen content plays an important role in guiding agricultural production activities. In this study, near-infrared hyperspectral imaging (NIR-HSI) combined with two spectral preprocessing algorithms, two characteristic wavelength selection algorithms and two machine learning algorithms were applied to determine the content of soil nitrogen. Two types of soils (laterite and loess, collected in 2020) and three types of nitrogen fertilizers, namely, ammonium bicarbonate (ammonium nitrogen, NH_4_-N), sodium nitrate (nitrate nitrogen, NO_3_-N) and urea (urea nitrogen, urea-N), were studied. The NIR characteristic peaks of three types of nitrogen were assigned and regression models were established. By comparing the model average performance indexes after 100 runs, the best model suitable for the detection of nitrogen in different types was obtained. For NH_4_-N, R^2^_p_ = 0.92, RMSE_P_ = 0.77% and RPD = 3.63; for NO_3_-N, R^2^_p_ = 0.92, RMSE_P_ = 0.74% and RPD = 4.17; for urea-N, R^2^_p_ = 0.96, RMSE_P_ = 0.57% and RPD = 5.24. It can therefore be concluded that HSI spectroscopy combined with multivariate models is suitable for the high-precision detection of various soil N in soils. This study provided a research basis for the development of precision agriculture in the future.

## 1. Introduction

Soil fertility is the ability of soil to supply and coordinate nutrients, water, air and heat for crop growth, and it is the most important indicator to measure the quality of soil resources [1]. Among the various nutrients in the soil, nitrogen (N), phosphorus (P) and potassium (K) are the necessary elements in large demand for crops, which are usually replenished by fertilization [2]. In particular, soil N, the main component of protein in crops, plays an important role in the growth of stems and roots and the development of fruits [3]. There are many types of soil nitrogen types. Among them, total nitrogen (TN) is the sum of various forms of nitrogen in the soil, representing the total storage capacity and nitrogen supply potential of soil nitrogen. TN can be divided into two categories: organic nitrogen and inorganic nitrogen. Organic nitrogen includes water-soluble nitrogen, hydrolyzed nitrogen and non-hydrolysable nitrogen; Inorganic nitrogen includes ammonium nitrogen, nitrate nitrogen and nitrous nitrogen. Available nitrogen (AN) refers to the nitrogen in the soil that could be easily absorbed and utilized by crops, and reflects the recent nitrogen supply capacity of the soil [4], mainly including ammonium nitrogen (NH_4_-N), nitrate nitrogen (NO_3_-N), amino nitrogen, amide nitrogen and some simple polypeptide and protein compounds. The application of N fertilizers is one of the most important management tools to ensure and increase yield in agricultural systems [5]. As common synthetic N fertilizers, NH_4_-N, NO_3_-N and amide nitrogen fertilizer all belong to the available nitrogen. The mechanism of action and effect on plants of different types of nitrogen fertilizers are also different. For example, NH_4_-N is easily soluble in water to produce ammonium ions and corresponding anions, which can be directly absorbed and utilized by crops with faster fertilizer efficiency as topdressing. NO_3_-N can promote the absorption of calcium, magnesium and potassium by crops, but it is easy to be lost with water, so it should not be used in paddy fields [6]. Amide nitrogen generally needs to be converted into ammonium nitrogen by soil microorganisms before being absorbed by crops. Therefore, the real-time determination of soil N content in different types is of great significance for agricultural production activities. In addition, the surplus and deficiency of soil nitrogen will change the quality and yield of crops to a certain extent [7]. Besides, the loss of nitrogen will lead to eutrophication of water bodies and the greenhouse effect [8]. Therefore, the effective and accurate measurement of soil nitrogen content has a guiding role in rational fertilization and precision agriculture.

The traditional methods of detecting N are mainly based on chemical analysis, such as the Kjeldahl method [9], the Dumas method [10], ultraviolet (UV) spectrophotometry [11], chromatography, etc. The chemical analysis method requires sample preparation in the early stage, which is time-consuming and labor-intensive, and is prone to human error. In addition, a large amount of strong acid and strong alkali needs to be used in the experimental process, and there are unsafe factors. Researchers have explored many rapid and nondestructive detection methods in nitrogen and found that the hyperspectral imaging (HSI) technique coupled with visible (vis) and/or near-infrared (NIR) spectroscopy is generally used for quantitative detection of fertilizer nitrogen, plant nitrogen and soil nitrogen, etc. [12]. For soil nitrogen detection, researchers have studied the effect of soil sample pretreatment methods on the NIR detection of TN, and found that the detection accuracy of soil after drying and sieving was better [13]. Moreover, results of detection accuracy are influenced by different machine learning algorithms and characteristic wavelengths selecting methods [14]. The main function of machine learning is to generate a model from empirical data on a computer by means of calculation, that is, a learning algorithm to achieve rapid judgment [15,16]. The machine learning model established based on spectral data can qualitatively or quantitatively detect nitrogen, phosphorus, potassium, pesticide residues and other substances in complex soil environments. For example, Kensuke Kawamura et al. used vis-NIR spectroscopy to estimate soil TN and found that the model using the characteristic wavelengths showed better prediction accuracy for TN than the full-band model [17]. Morellos et al. explored NIR spectroscopy combined with different machine learning algorithms to predict soil TN and found that the Cubist model provided the best prediction results [18]. Xu et al. used HSI combined with machine learning algorithms to detect TN, AN, NO_3_-N and NH_4_-N in soil profiles and found that SVM model performed best [19]. Previous studies have shown the potential of using NIR or HSI coupled with multivariate data analysis for the detection of soil nitrogen content in soil. In addition, the detection speed of HSI is faster than that of point-based techniques, as many samples can be scanned and analyzed at the same time by using an HSI camera [20]. However, to the best of our knowledge, many studies only focused on the detection of TN content, and so far, no study has been carried out to detect the NH_4_-N, NO_3_-N and urea-N in different types of soils by using HSI simultaneously, so it is urgent to fulfill the research gaps in these aspects. Moreover, we also want to further explore the possibility of improving the accuracy of soil nitrogen detection.

In view of the shortcomings of existing research studies, this paper mainly explored 2 problems: (1) to explore the spectral responses of different nitrogen fertilizers and different kinds of soils in the NIR band, and to illustrate the feasibility of detection in principle. (2) to compare the impact of different machine learning algorithms and characteristic wavelength selection algorithms on soil nitrogen detection, and to select the optimal model. It was hypothesized that (1) laboratory-based HSI spectroscopy would be capable of capturing the spectral properties of low nitrogen levels in soils, and (2) HSI spectroscopy combined with machine learning techniques could also perform well for simultaneously predicting the various soil N at the fine scale. To test our hypotheses, 2 soils with different properties from different areas were collected and 3 nitrogen fertilizers were purchased. The focus of this research includes the following parts: (1) Analyze the NIR reflectance spectral characteristics of 2 soils and 3 nitrogen fertilizer standards (ammonium bicarbonate, sodium nitrate and urea); (2) Compare the detection results of 2 preprocessing algorithms and 2 machine learning models in different types of nitrogen fertilizers using the full-band spectra; (3) Use 2 characteristic wavelength selection algorithms to select the characteristic wavelength on the spectra processed by the optimal preprocessing method, and compare them with the full-band models to obtain the optimal models. This research aimed to provide theoretical and experimental guidance for the realization of precision fertilization.

## 2. Results and Discussion

### 2.1. Soil Chemical Properties Analysis

Due to the complex soil composition, the organic matter and water in the soil all respond to the NIR spectra, so before analyzing the spectral characteristics of soils with different nitrogen contents, the main chemical composition values of the collected soils were measured and analyzed. It can be seen from Table 1 that soil 1 is acidic soil (pH = 4.69), and soil 2 is alkaline soil (pH = 8.85); the conductivity of soil 1 (44.3 μm/cm) is much lower than that of soil 2 (346 μm/cm); the available nitrogen of soil 1 (31.45 mg/kg) was slightly lower than that in soil 2 (42.19 mg/kg); the content of available potassium (8.595 mg/kg) and available phosphorus (1.45 mg/kg) in soil 1 was much lower than that in soil 2; the organic matter in soil 1 (0.59%) has a content slightly lower than soil 2 (0.63%). Overall, the chemical compositions of the 2 soils were quite different.

### 2.2. Soil Spectral Feature Analysis

Figure 1a shows the NIR spectral reflectance curves of the 2 soils. It was observed that both soils have absorption peaks around 1410 nm, which could be assigned to the O-H first overtones. The crystal water peak of minerals was reflected here [21], and the absorption peak of soil 1 was more obvious than that of soil 2, so its mineral content was higher.

The reflectance of soil2 is lower than that of soil1 owning to the higher organic matter content of soil2 than that of soil1, which has been confirmed in previous studies [22].

Figure 1b shows the NIR spectral reflectance curves of three nitrogen fertilizer standards. For the three nitrogen fertilizer standards, there were clear differences in the spectra between 900 and 1700 nm. The specific performance was that NH_4_-N, NO_3_-N had no obvious absorption peaks, while urea-N had multiple obvious absorption peaks. According to relevant literature [23], the absorption peak of urea molecule at 1160 nm could be assigned to the C=O stretch fourth overtone, and the absorption peaks at 1460, 1490 and 1520 nm could be ascribed to the Sym N−H stretch first overtone, Sym N−H stretch first overtone and N−H stretch first overtone, respectively.

Figure 2 shows the average NIR reflectance spectra of soil sample sets at different nitrogen content levels. Combining with Figure 1, the absorption peaks of the 3 nitrogen fertilizers themselves were not reflected in the waveform diagram, which may be submerged due to the low concentration.

Observing the waveform of each sample set, it was obvious that with the change of nitrogen concentration, the spectral reflectance also changes, which showed that there was a certain correlation between nitrogen concentration and reflectance. It is concluded that NIR spectra had spectral responses to soil nitrogen concentrations. In order to further explore the relationship between spectral reflectance and nitrogen concentration, data mining methods should be used to analyze the high-dimensional data. Due to the obvious noise at the beginning and end of the spectra, the 975–1645 nm bands with high signal-to-noise ratio were selected for the subsequent data analysis.

### 2.3. Models Based on Full-Wavelength Spectra

Taking the reflectance spectra as the independent variables and the 12 nitrogen concentrations of the soil as the dependent variables, the PLSR and LSSVM models were established using the raw spectra, multiplicative scatter correction (MSC), and wavelet transform (WT) preprocessed spectra, respectively. Table 2 shows the average results of 100 runs of the models under the optimal preprocessing method, standard deviation (SD) is shown in parentheses. For soil1_NH_4_-N, the results after MSC preprocessing were the best, for soil2_NH4-N, soil2_NO3-N, soil1_urea-N and soil2_urea-N, the results after WT preprocessing were the best, and for soil1_NO_3_ -N, the raw spectra got satisfactory results.

In terms of model performance, the RPD of the PLSR and LSSVM models for the 6 data sets were both above 2, indicating that the models had reliable prediction accuracy. Among them, the results of soil urea-N were the best. In soil1, the prediction set R^2^_p_ of PLSR and LSSVM reached 0.94 and 0.91, respectively, and RMSE_P_ were 0.66% and 0.75%, respectively. In soil2, the R^2^_p_ of both PLSR and LSSVM was 0.92, and the RMSE_P_ was 0.79% and 0.73%, respectively. For NH_4_-N, in soil1, the LSSVM (R^2^_p_ = 0.86, RMSE_P_ = 1.07%) model outperformed PLSR, while in soil2, PLSR (R^2^_p_ = 0.88, RMSE_P_ = 0.98%) performed significantly better than LSSVM (R^2^_p_ = 0.80, RMSE_P_ = 1.21%). For NO_3_-N, PLSR performed slightly better than LSSVM in both soil1 and soil2, with R^2^_p_ of 0.88 and 0.78, and RMSE_P_ of 1.08% and 1.31%, respectively.

In summary, NIR-HSI combined with PLSR and LSSVM algorithms could effectively detect nitrogen content in soils. The predicted results of urea-N were better than that of NH_4_-N and NO_3_-N; this might be related to the obvious characteristic peaks of urea molecules in NIR band.

### 2.4. Characteristic Wavelengths Selection

The full-band-based NIR spectral dataset contains 200 wavelengths of data, the amount of data was large and there was a lot of redundant information irrelevant to the spectral response of soil nitrogen. Therefore, competitive adaptive reweighted sampling (CARS) and successive projections algorithm (SPA) feature wavelength selection algorithms were used to find the feature wavelengths related to soil nitrogen content. The number of characteristic wavelengths selected from the 6 data sets and the ratio to the number of full wavelengths are shown in Table 3.

Overall, after the selection of characteristic wavelengths, variable numbers were greatly reduced in the 6 datasets. The wavelengths selected by the CARS were between 16 and 44, accounting for 8–24.5% of the full band, and the wavelengths selected by the SPA were between 8 and 14, accounting for 5–7% of the full band. On each dataset, there were fewer characteristic wavelengths selected by SPA than by CARS. Among them, for soil1_urea-N and soil2_urea-N, the characteristic wavelengths selected by CARS were reduced to 8% and by the SPA, 5%, respectively.

In order to show the positions of the characteristic wavelengths more intuitively, we took the first-order derivation of the average spectra of the 6 data sets, and marked the CARS and SPA methods with vertical lines of different colors in Figure 3. The peaks and troughs of the first-order derivative spectra show the differences between the spectra of different nitrogen concentrations. On the whole, whether it was CARS or SPA, the selected characteristic wavelengths basically covered the positions where these differences appeared, and the positions where the 2 appear overlap to some extent. Specifically, for the same type of nitrogen, in the soil1 dataset, the spectra had a significant difference around 1400 nm, and the characteristic wavelengths selected by the two algorithms were mainly concentrated here; the differences at 1000 nm and 1600 nm were not obvious, so fewer characteristic wavelengths were covered. For soil2, the overall differences of the spectra were not as obvious as that of soil1. Although the selected characteristic wavelengths were also basically around 1000, 1400 and 1600 nm, the numbers of selected characteristic wavelengths were not as large as those of soil1. In addition, for different types of nitrogen in the same soil, the positions and quantities of characteristic wavelengths were obviously different.

Therefore, the characteristic wavelengths founded by the CARS and SPA methods were closely related to the nitrogen content in the soil. However, in previous literatures [24,25], the characteristic wavelengths of soil nitrogen were rarely selected, and even if the characteristic wavelength selection algorithm was used, the location of their bands was not analyzed specifically or visually. In order to prove whether the selected wavelengths were reliable, further analysis through machine learning modeling was conducted.

### 2.5. Models Based on Characteristic Wavelengths

Table 4 shows the PLSR and LSSVM model performances based on characteristic wavelengths selected by CARS. For soil1_NH_4_-N and soil2_NH_4_-N, the results of PLSR were better than LSSVM, with R^2^_p_ reaching 0.93 and 0.91, respectively, which were improved by 0.07 and 0.03, respectively, compared with full-band modeling; similarly, for NO_3_-N, the results of CARS-PLSR were also better. The R^2^_p_ in soil1 and soil2 were 0.96 and 0.88, respectively, which were 0.12 and 0.10 higher than before, and the increases were more obvious. Among the 3 types of nitrogen, the detection result of urea-N was the best. In soil1_urea-N, the R^2^_p_ (0.96) of the CARS-PLSR and CARS-LSSVM models were consistent, which was 0.02 higher than that of the full-band model, but the RMSE_P_ (0.53) of CARS-LSSVM was lower and the RPD (5.65) was higher, so its performance was slightly better. In soil1_urea-N, the result of CARS-PLSR (R^2^_p_ = 0.95) was better, which was 0.03 higher than before.

Table 5 shows the detection results of soil nitrogen content based on the SPA-PLSR and SPA-LSSVM models. For NH_4_-N, the best result on soil1 (R^2^_p_ = 0.87) was only 0.01 higher than that of the full-band model; the best result on soil2 (R^2^_p_ = 0.86) was 0.02 lower than the full-band model. It can be seen from Table 5 that the number of characteristic wavelengths selected by SPA in this dataset was only 4% of the full band. Although the redundancy of spectral information was greatly reduced, it might make some useful features related to NH_4_-N in soil2 lost. On the soil1_NO_3_-N and soil2_NO_3_-N datasets, the results that performed better were all SPA-PLSR, with R^2^_p_ of 0.89 and 0.84, respectively. Similarly, urea-N got the best predicted results, SPA-PLSR performed slightly better on both soil1 (R^2^_p_ = 0.95, RMSE_P_ = 0.62, RPD = 4.43) and soil2 (R^2^_p_ = 0.96, RMSE_P_ = 0.60, RPD = 4.82) datasets. Compared with the full-band models, the improvements were 0.01 and 0.04, respectively.

Combining Table 4 and Table 5, after selecting characteristic wavelengths by CARS and SPA, the prediction accuracy of the models on most datasets had been improved to some extent, and the results on only a few datasets were similar or slightly reduced.

### 2.6. Predictive Fit of the Optimal Model

By comprehensively comparing the average R^2^_p_, RMSE_P_ and RPD of all models, the best model suitable for detecting different types of nitrogen and soils could be obtained. On the soil1_NH_4_-N, soil2_NH_4_-N, soil1_NO_3_-N, soil2_NO_3_-N, soil1_urea-N and soil2_urea-N datasets, the models with the best prediction results were CARS-PLSR, CARS-PLSR, CARS-PLSR, CARS-PLSR, CARS-LSSVM and SPA-PLSR, respectively. We output the results of the best models on the prediction sets and fitted them with scatter plots, as shown in Figure 4.

For the detection of NH_4_-N in the 2 soils, the average R^2^_p_ of the CARS-PLSR model was 0.92, the average RMSE_P_ was 0.77% and the average RPD was 3.63, which indicated that the model proposed in this study could make reliable prediction when the NH_4_-N content in soil was higher than 0.77%. Compared with the result of Shengxiang Xu et al. [19] using the SVMR model (R^2^_p_ = 0.70), this study improved by 0.22. In the datasets of soil1_NO_3_-N and soil2_NO_3_-N, the average R^2^_p_ of CARS-PLSR was 0.92, the average RMSE_P_ was 0.74%, and the average RPD was 4.17, which meant that the NO_3_-N content in the soil was higher than 0.74% and the model could make accurate prediction. Shengxiang Xu et al. [19] used the SVMR model to detect NO_3_-N with a R^2^_p_ of 0.82, which was improved by 0.1 in this study. For urea-N, the models had an average R^2^_p_ of 0.96 in both soils, an average RMSE_P_ of 0.57%, and an average RPD of 5.24. That is, when the urea-N content in the soil was higher than 0.57%, the model could achieve a good prediction effect. The R^2^_p_ reached by Yong He et al. [25] using the PLS model was 0.94, which was improved by 0.02 in this study. Comparing the 3 types of nitrogen, it was found that the detection effect of the model in NH_4_-N and NO_3_-N was very close, and the prediction ability for urea-N was significantly better, which was shown in Figure 4e,f as a good fit of the actual and predicted values on both sides of the trend line. Moreover, the detection accuracy of the 3 types of nitrogen detected by this method had been improved compared with the previous literatures, and the PLSR model was found to be more stable than the LS-SVM model.

## 3. Materials and Methods

### 3.1. Sample Preparation

The surface soil samples (0–20 cm) were collected from two regions in China, named soil1 and soil2, respectively. Soil1 is Fujian laterite, and soil2 is Shanxi loess. They were dried in an indoor ventilated environment. In order to understand the main chemical properties of the two soil samples collected [26], a part of each of the soil samples was used for chemical property detection. The pH value was measured by a pH meter (LEICI PHSJ-4ApH, Shanghai Kuosi Electronics Co., Ltd., Shanghai, China); the conductivity was measured by a conductivity meter (LEICI DDSJ-318, Shanghai Kuosi Electronics Co., Ltd., Shanghai, China); the organic matter was measured by potassium dichromate oxidation-spectrophotometry; the available nitrogen was measured by the alkaline hydrolysis diffusion method, available phosphorus was determined by sodium bicarbonate leaching-molybdenum antimony anti-spectrophotometry, and available potassium was determined by flame atomic absorption spectrometry.

In this study, 3 representative nitrogen fertilizer standards, including ammonium bicarbonate (Aladdin, AR, Hangzhou Hehui Chemical Co., Ltd., Hangzhou, China), sodium nitrate (AR, ≥99.0%, Sinopharm Group Chemical Reagent Co., Ltd., Shanghai, China) and urea (Aladdin, AR, 99%, Tianjin Zhiyuan Chemical Reagent Co., Ltd., Tianjin, China) were used. The 3 types of nitrogen fertilizer solutions were mixed with 2 kinds of soil, and a total of 6 soil sample sets were obtained, named soil1_NH_4_-N, soil2_NH_4_-N, soil1_NO_3_-N, soil 2_NO_3_-N, soil1_urea-N, soil2_urea-N. Each soil sample set contains 12 concentration gradients, and the nitrogen content of them are shown in Table 6. The mixed soil was naturally air-dried at room temperature, sieved through a 100-mesh sieve after grinding, and a tableting machine (TUOPU FW-4A, Tianjin Sichuang Jingshi Technology Development Co., Ltd., Tianjin, China) was used to apply a pressure of 1000 MPa to the soil so that each sample obtained a weight of about 0.3 g, a thickness of about 1 mm, and a diameter of about 13 mm, as shown in Figure 5. Among them, 10 parallel samples were set for each gradient, and a total of 120 (12 × 10 = 120) samples were set for each sample set. In addition, in order to explore the spectral characteristics of nitrogen fertilizers, the 3 nitrogen fertilizer standards were compressed respectively, and the tableting procedure is the same as for soil.

### 3.2. Hyperspectral Image Acquisition

The core components of the NIR-HSI system consist of a high-performance CCD (Charge-Coupled Device Camera), a mobile stage for moving scanning of the sample, an imaging spectrometer (ImspectorV10, Spectral Imaging Ltd., Oulu, Finland) and image acquisition software. A lighting fixture is mounted above the mobile platform, containing 2150 W quartz tungsten halogen lamps. A lab-based line scan HSI system to is used to control the hyperspectral image acquisition process, including the speed of the moving stage motors, exposure time, and wavelength range. Hyperspectral images of the prepared samples were obtained in the 870–1740 nm wavelength range and reflectance mode. Finally, a total of 6 soil hyperspectral images and 1 nitrogen fertilizer standard image were obtained. To obtain higher-quality images, it is necessary to correct the reflectivity of the original image ($I_{0}$) [27]. In the same environment as the sample image acquisition, the standard white calibration plate (reflectivity close to 100%) was scanned to obtain a white calibration image (W), then turn off the light source and cover the lens cover (reflectivity close to 0%) to collect a black calibration image (B). The corrected images (I) were calculated according to the following formula [28]:(1)$$I=\frac{I_{0}-B}{W-B}\times 100\%$$

All the corrected images were then used as the basis for subsequent analysis to then extract the regions of samples from the background and calculate the average spectrum of each sample.

### 3.3. Data Analysis

#### 3.3.1. Data Preprocessing

Two preprocessing methods were used in this study to correct the spectral data, including the wavelet transform (WT) and multiplicative scatter correction (MSC).

WT is a time–frequency analysis method, which is mainly used in the research of spectral data compression and denoising [29]. It has the characteristics of multiresolution, low entropy, flexible selection of basic functions and de-correlation, high resolution in the high-frequency range, and low-frequency resolution in the low-frequency range, so it is suitable for analyzing signals of any scale.

The principle of the MSC method is to obtain an ideal spectrum by performing scattering correction on each wavelength point, and to correct and reduce the spectral variation difference caused by light scattering on the surface of an inhomogeneous sample. It is a common spectral preprocessing method used to correct the dispersion effect of the spectrum and reduce the occurrence of spectral baseline drift [30].

#### 3.3.2. Spectral Feature Selection

Hyperspectral images could provide a large amount of spectral and spatial information related to the vitality properties of the soil nitrogen; however, they also contain overlapping and redundant information, so it is necessary to use feature selection algorithms to obtain representative and important wavelengths that can reduce irrelevant information and increase computational speed.

The principle of competitive adaptive reweighted sampling (CARS) algorithm is to simulate the principle of Darwinian evolution theory (survival of the fittest). In each sampling process, the combination of adaptive re-weighted sampling technology (ARS) and exponential decay function is used to select the wavelength of the regression coefficient with a large absolute value in the PLS model, and remove the wavelength with a smaller weight of the regression coefficient value. Based on the tenfold cross-validation, the subset with the smallest root mean square error in the prediction of the PLS subset model is selected, and this subset is the optimal variable combination. After N sampling, N variable subsets are obtained, and the variable subset with the smallest cross-validation root mean square error (RMSECV) is selected according to cross-validation, and the variables contained in this subset are the optimal characteristic wavelength variable combination [31,32].

The successive projections algorithm (SPA) is a forward feature variable selection method. It projects the vector and compares the size of the projection vector by projecting the wavelength to other wavelengths. The wavelength with the largest projection vector is the wavelength to be selected, and then the final characteristic wavelength is selected based on the correction model. The SPA selects the combination of NIR spectral variables with the least redundant information and the least collinearity [33].

#### 3.3.3. Model Building and Analysis

The main function of machine learning (ML) is to generate a model from empirical data on a computer by means of calculation, that is, a learning algorithm to achieve rapid judgment. The ML model established based on spectral data can qualitatively or quantitatively detect organic matter content, pesticide residues, heavy metal pollution, etc., in complex soil environments. The modeling method in this study is mainly used to establish a rapid detection model of nitrogen content in soils.

Partial least squares regression (PLSR) is a commonly used multivariate regression modeling method [34,35]. In this study, the PLSR algorithm established a linear regression prediction model by projecting the soil nitrogen content y and the corresponding spectral response matrix *x* into a new space. In the process of establishing the model, the PLSR algorithm extracted the principal components N of *y* and *x,* and the maximum extract is the correlation between the principal components from *y* and *x*, and find the multidimensional direction of the x space to explain the multidimensional direction with the largest variance in the y space [36].

The least squares support vector machine (LSSVM) algorithm is an improved regression method based on the SVM algorithm [37], and it has been widely used in previous research studies [38,39]. In this study, we constructed a regression model with spectral data x as the independent variables and nitrogen content y as the dependent variables. The LSSVM regression algorithm maps the input data from the normal space to the high-dimensional space, replaces the inequality constraints with equality constraints, and solves the minimum loss function in the high-dimensional space to obtain a linear fitting equation [40]. Given the training dataset $D=\left\{\left(x_{1},y_{1}\right),\left(x_{2},y_{2}\right),\ldots ,\left(x_{i},y_{i}\right),x_{i}\in R,y_{i}\in R,\right\}$, we built a linear prediction model, as shown in Equation (2), dividing in a ratio of 0.7 to the training set and dataset randomly. In the dataset, $x_{i}$ and $y_{i}$ denote respectively the spectral signature and N fertilizer content of the ith soil sample. In Equation (2), ω represents the weight vector, and b represents the offset value.
(2)$$y(x)=\omega^{T}x+b$$

The main data analysis software are ENVI4.6 software (ITT Visual Information Solutions, Boulder, CO, USA) and MATLAB 2020b software (Mathworks Inc., Natick, MA, USA). The flowchart of data acquisition and processing flow of this study is shown in Figure 6.

#### 3.3.4. Model Performance Evaluation

The regression model evaluates the detection performance of the established model through the coefficient of determination (R^2^), the root mean square error (RMSE) and the relative prediction deviation (RPD). In this paper, the R^2^ reflects the interpretation ability of the constructed model for sample spectra, RMSE refers to the errors between the predicted and the actual nitrogen content, and RPD evaluates the overall performance of the model. In general, the closer the R^2^ value is to 1, the closer the RMSE value is to 0, and the larger the RPD value, the better the performance of the model [41].

## 4. Conclusions

In this study, the NIR-HSI instrument was used to detect two kinds of soils after tableting (latter soil and loess) added with NH_4_-N, NO_3_-N and urea-N solutions. Machine learning models were established based on the full band and the characteristic bands respectively, and the optimal prediction models for different nitrogen detection were obtained by comprehensively comparing the evaluation indicators. First, we compared the results of PLSR and LSSVM modeling of spectra with no preprocessing, MSC and WT methods, and obtained the most suitable pretreatment method for the detection of different types nitrogen in different soil regions, which was used for subsequent data analysis. Second, this study significantly reduced the spectral dimension required for modeling by using two characteristic wavelength selection algorithms. It was found that the numbers and positions of characteristic wavelengths in different datasets were significantly different, while the wavelengths selected by different algorithms had a certain degree of similarity. Finally, the optimal model was obtained by comprehensively analyzing the performances of different models on different datasets and the average results were calculated. In NH_4_-N, R^2^_p_ was 0.92, RMSEP was 0.77% and RPD was 3.63; for NO_3_-N, R^2^_p_ was 0.92, RMSE_P_ was 0.74% and RPD was 4.17; for urea-N, R^2^_p_ was 0.96, RMSE_P_ was 0.57% with RPD of 5.24.

## Figures and Tables

**Figure 1 molecules-27-02017-f001:**
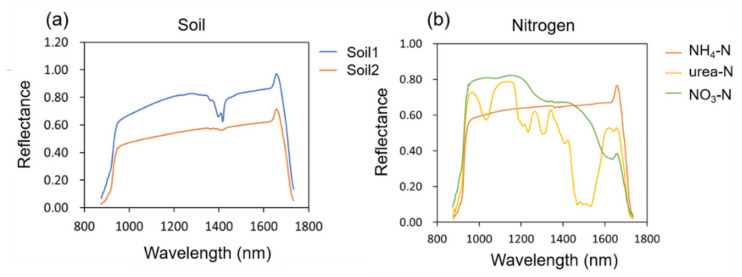
Spectra of soils and nitrogen fertilizer standards: (**a**) Spectra of soil1 and soil2; (**b**) Spectra of nitrogen fertilizer standards.

**Figure 2 molecules-27-02017-f002:**
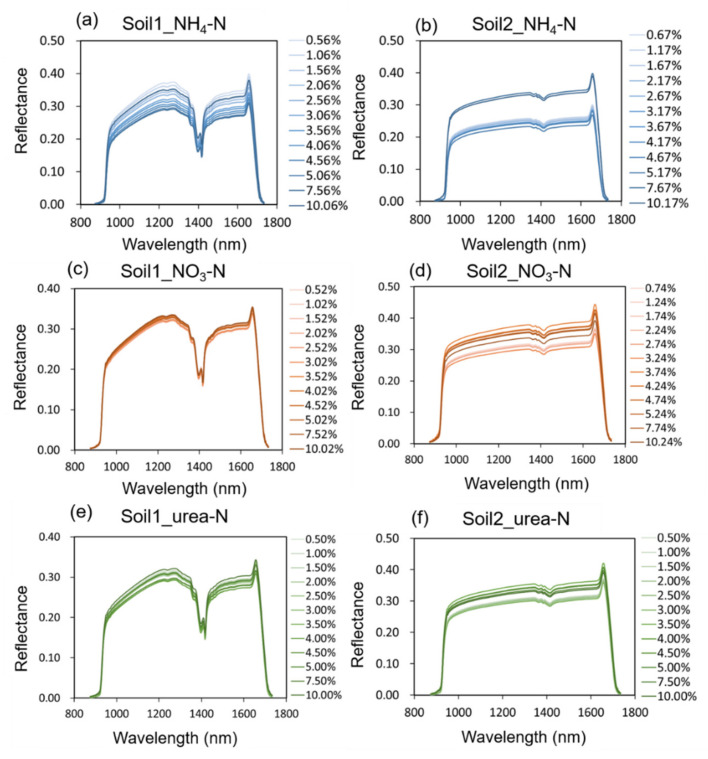
Average NIR reflectance spectra of the six sample sets: (**a**) Soil1_NH_4_-N; (**b**) Soil2_NH_4_-N; (**c**) Soil1_NO_3_-N; (**d**) Soil2_NO_3_-N; (**e**) Soil1_urea-N; (**f**) Soil2_urea-N.

**Figure 3 molecules-27-02017-f003:**
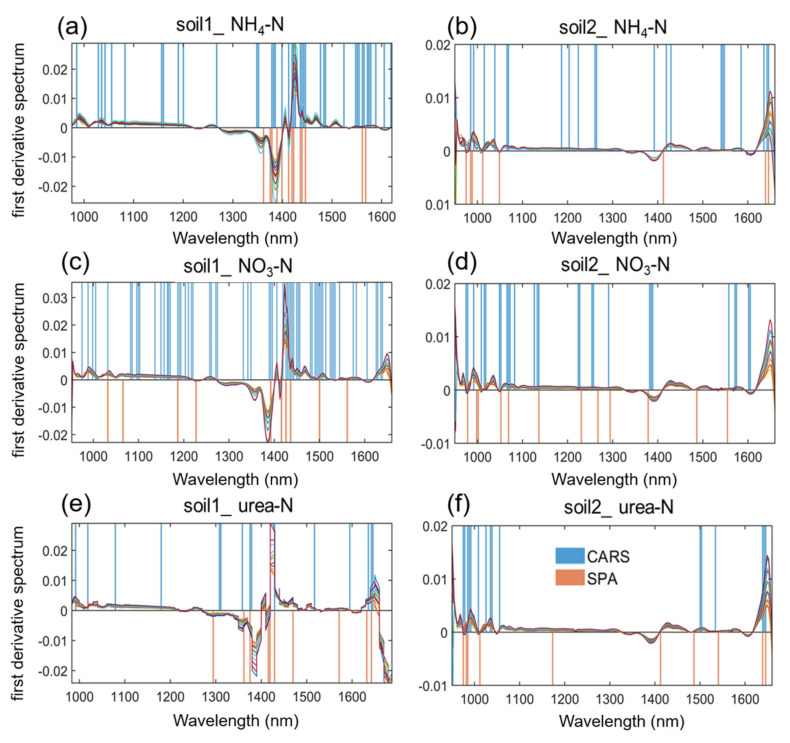
The positions of the characteristic wavelengths on the first derivative spectra: (**a**) Soil1_NH_4_-N; (**b**) Soil2_NH_4_-N; (**c**) Soil1_NO_3_-N; (**d**) Soil2_NO_3_-N; (**e**) Soil1_urea-N; (**f**) Soil2_urea-N.

**Figure 4 molecules-27-02017-f004:**
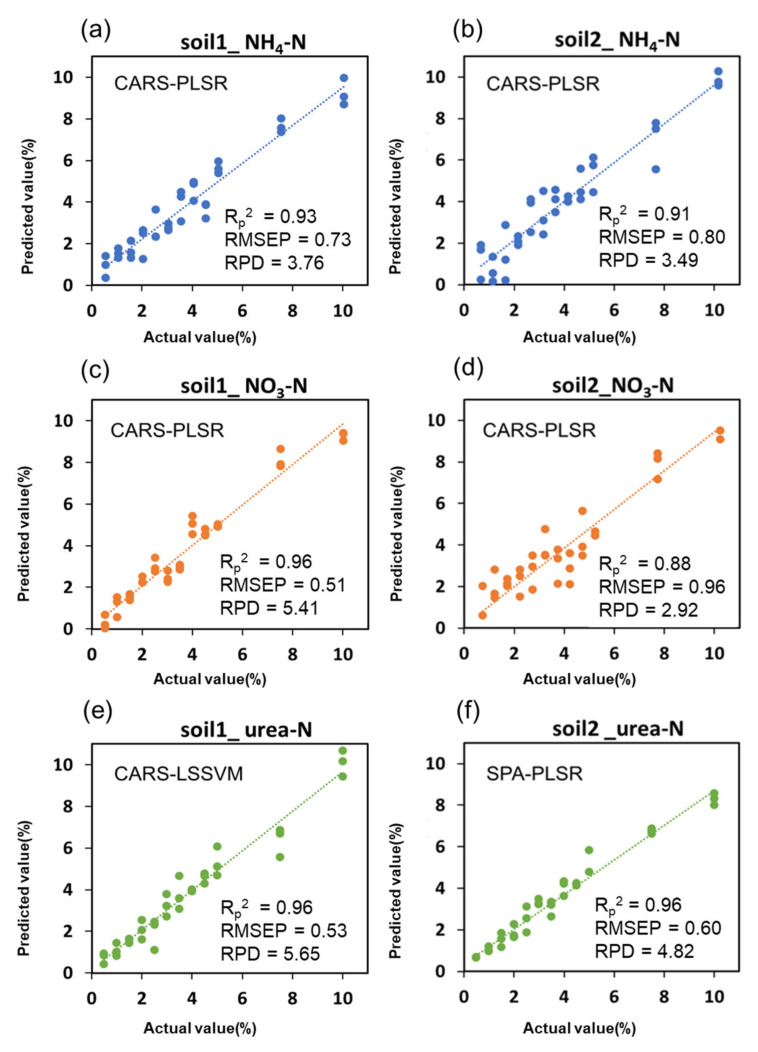
Plots of actual values versus predicted values of soil nitrogen content: (**a**) Soil1_NH_4_-N; (**b**) Soil2_NH_4_-N; (**c**) Soil1_NO_3_-N; (**d**) Soil2_NO_3_-N; (**e**) Soil1_urea-N; (**f**) Soil2_urea-N.

**Figure 5 molecules-27-02017-f005:**
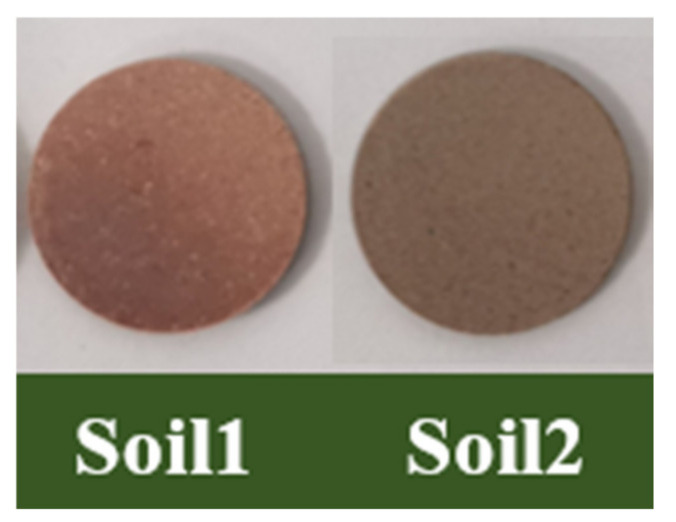
The RGB image of soil samples after tableting.

**Figure 6 molecules-27-02017-f006:**
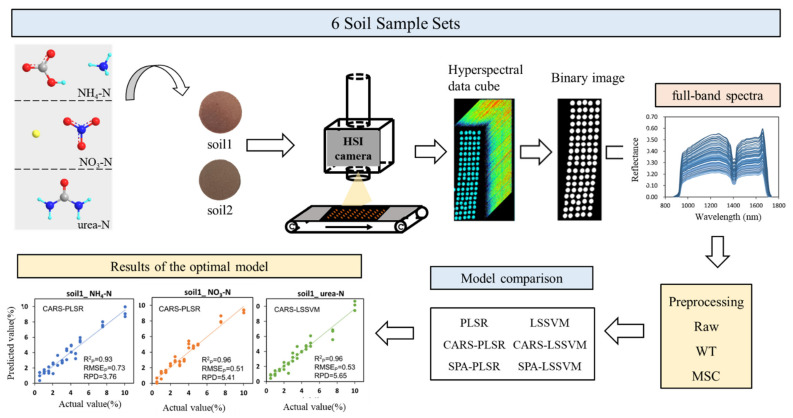
Flowchart of image collection and data analysis.

**Table 1 molecules-27-02017-t001:** The chemical properties of soil samples.

Soil Type	pH	Electrical Conductivity (μm/cm)	Available Nitrogen (mg/kg)	Available Potassium (mg/kg)	Available Phosphorus (mg/kg)	Organic Matter (%)
Soil1	4.69	44.3	31.45	8.60	1.45	0.59
Soil2	8.85	346	42.19	265.88	14.48	0.63

**Table 2 molecules-27-02017-t002:** Model performances (mean with SD in parentheses) of full-wavelength based on different preprocessing methods.

Dataset	Preprocessing	Model	R^2^_C_	RMSE_C_ (%)	R^2^_P_	RMSE_P_ (%)	RPD
soil1_NH_4_-N	MSC ^1^	PLSR ^4^	0.97(0.02)	0.41(0.27)	0.84(0.04)	1.07(0.14)	2.57(0.31)
		LSSVM ^5^	0.97(0.01)	0.43(0.16)	0.86(0.04)	1.01(0.13)	2.70(0.30)
soil2_NH_4_-N	WT ^2^	PLSR	0.93(0.01)	0.71(0.07)	0.88(0.03)	0.98(0.12)	2.91(0.38)
		LSSVM	0.94(0.05)	0.57(0.30)	0.80(0.08)	1.21(0.27)	2.40(0.53)
soil1_NO_3_-N	Raw ^3^	PLSR	0.93(0.03)	0.69(0.20)	0.84(0.04)	1.08(0.14)	2.57(0.36)
		LSSVM	0.99(0.03)	0.24(0.23)	0.83(0.08)	1.10(0.25)	2.58(0.50)
soil2_NO_3_-N	WT	PLSR	0.90(0.04)	0.80(0.16)	0.78(0.05)	1.31(0.21)	2.19(0.25)
		LSSVM	0.92(0.05)	0.71(0.23)	0.73(0.09)	1.43(0.30)	2.02(0.31)
soil1_urea-N	WT	PLSR	0.97(0.01)	0.47(0.05)	0.94(0.01)	0.66(0.08)	4.19(0.51)
		LSSVM	0.98(0.02)	0.37(0.15)	0.91(0.11)	0.75(0.38)	4.32(1.33)
soil2_urea-N	WT	PLSR	0.97(0.02)	0.47(0.12)	0.92(0.03)	0.79(0.12)	3.66(0.57)
		LSSVM	0.98(0.01)	0.35(0.12)	0.92(0.06)	0.73(0.25)	4.24(1.29)

^1^ MSC: multiplicative scatter correction; ^2^ WT: wavelet transform; ^3^ Raw: raw data; ^4^ PLSR: partial least squares regression; ^5^ LSSVM: least squares support vector machine.

**Table 3 molecules-27-02017-t003:** Characteristic wavelength selection results based on CASR and SPA.

Dataset	Method	Variable Number	Proportion
soil1_NH_4_-N	CARS ^1^	44	22%
	SPA ^2^	14	7%
soil2_NH_4_-N	CARS	21	10.5%
	SPA	8	4%
soil1_NO_3_-N	CARS	49	24.5%
	SPA	10	5%
soil2_NO_3_-N	CARS	28	14%
	SPA	12	6%
soil1_urea-N	CARS	16	8%
	SPA	10	5%
soil2_urea-N	CARS	16	8%
	SPA	10	5%

^1^ CARS: adaptive reweighted sampling; ^2^ SPA: successive projections algorithm.

**Table 4 molecules-27-02017-t004:** PLSR and LSSVM model performances based on characteristic wavelengths selected by CARS.

Dataset	Model	R^2^_C_ ^3^	RMSE_C_ ^4^ (%)	R^2^_P_ ^5^	RMSE_P_ ^6^ (%)	RPD ^7^
soil1_NH_4_-N	CARS-PLSR ^1^	0.96(0.00)	0.53(0.03)	0.93(0.01)	0.73(0.06)	3.76(0.34)
	CARS-LSSVM ^2^	0.97(0.01)	0.47(0.11)	0.90(0.04)	0.84(0.15)	3.32(0.49)
soil2_NH_4_-N	CARS-PLSR	0.93(0.01)	0.69(0.04)	0.91(0.02)	0.80(0.10)	3.49(0.44)
	CARS-LSSVM	0.95(0.04)	0.56(0.22)	0.82(0.10)	1.17(0.60)	2.64(0.71)
soil1_NO_3_-N	CARS-PLSR	0.98(0.00)	0.34(0.03)	0.96(0.01)	0.51(0.06)	5.41(0.68)
	CARS-LSSVM	0.98(0.01)	0.30(0.06)	0.95(0.06)	0.60(0.22)	4.83(0.98)
soil2_NO_3_-N	CARS-PLSR	0.92(0.01)	0.75(0.03)	0.88(0.02)	0.96(0.10)	2.92(0.27)
	CARS-LSSVM	0.94(0.03)	0.62(0.14)	0.76(0.09)	1.33(0.27)	2.17(0.40)
soil1_urea-N	CARS-PLSR	0.97(0.00)	0.45(0.02)	0.96(0.01)	0.54(0.05)	5.05(0.45)
	CARS-LSSVM	0.98(0.01)	0.35(0.06)	0.96(0.03)	0.53(0.14)	5.65(1.24)
soil2_urea-N	CARS-PLSR	0.97(0.00)	0.47(0.03)	0.95(0.01)	0.59(0.07)	4.76(0.54)
	CARS-LSSVM	0.98(0.01)	0.31(0.12)	0.92(0.08)	0.75(0.29)	4.16(1.19)

^1^ CARS-PLSR: adaptive reweighted sampling—partial least squares regression; ^2^ CARS-LSSVM: adaptive reweighted sampling—least squares support vector machine;^3^ R^2^_C_: coefficient of determination of calibration; ^4^ RMSE_c_: root mean square error of calibration; ^5^ R^2^_P_: coefficient of determination of prediction; ^6^ RMSE_p_: root mean square error of prediction;^7^ RPD: relative prediction deviation.

**Table 6 molecules-27-02017-t006:** Soil nitrogen content of 6 sample sets.

Sample Set	Content of Nitrogen (%)	Number of Samples
Soil1_NH_4_-N	0.56, 1.06, 1.56, 2.06, 2.56, 3.06, 3.56, 4.06, 4.56, 5.06, 7.56, 10.06	120
Soil2_NH_4_-N	0.67, 1.17, 1.67, 2.17, 2.67, 3.17, 3.67, 4.17, 4.67, 5.17, 7.67, 10.17	120
Soil1_NO_3_-N	0.52, 1.02, 1.52, 2.02, 2.52, 3.02, 3.52, 4.02, 4.52, 5.02, 7.52, 10.02	120
Soil2_NO_3_-N	0.74, 1.24, 1.74, 2.24, 2.74, 3.24, 3.74, 4.24, 4.74, 5.24, 7.74, 10.24	120
Soil1_urea-N	0.50, 1.00, 1.50, 2.00, 2.50, 3.00, 3.50, 4.00, 4.50, 5.00, 7.50, 10.00	120
Soil2_urea-N	0.50, 1.00, 1.50, 2.00, 2.50, 3.00, 3.50, 4.00, 4.50, 5.00, 7.50, 10.00	120

**Table 5 molecules-27-02017-t005:** PLSR and LSSVM model performances based on characteristic wavelengths selected by SPA.

Dataset	Model	R^2^_C_	RMSE_C_ (%)	R^2^_P_	RMSE_P_ (%)	RPD
soil1_NH_4_-N	SPA-PLSR ^1^	0.90(0.01)	0.81(0.03)	0.87(0.02)	0.96(0.08)	2.85(0.25)
	SPA-LSSVM ^2^	0.92(0.02)	0.76(0.07)	0.87(0.02)	0.99(0.08)	2.77(0.08)
soil2_NH_4_-N	SPA-PLSR	0.87(0.01)	0.93(0.06)	0.86(0.04)	1.03(0.15)	2.72(0.43)
	SPA-LSSVM	0.92(0.04)	0.71(0.20)	0.79(0.09)	1.26(0.28)	2.27(0.41)
soil1_NO_3_-N	SPA-PLSR	0.91(0.01)	0.81(0.04)	0.89(0.02)	0.92(0.09)	2.99(0.26)
	SPA-LSSVM	0.92(0.01)	0.75(0.06)	0.86(0.06)	1.00(0.22)	2.83(0.39)
soil2_NO_3_-N	SPA-PLSR	0.87(0.01)	0.86(0.04)	0.84(0.03)	1.11(0.09)	2.49(0.21)
	SPA-LSSVM	0.92(0.04)	0.73(0.20)	0.73(0.10)	1.42(0.27)	2.02(0.35)
soil1_urea-N	SPA-PLSR	0.96(0.00)	0.56(0.03)	0.95(0.01)	0.62(0.06)	4.43(0.50)
	SPA-LSSVM	0.97(0.04)	0.46(0.17)	0.92(0.14)	0.70(0.52)	4.56(1.07)
soil2_urea-N	SPA-PLSR	0.96(0.00)	0.51(0.03)	0.96(0.01)	0.60(0.06)	4.82(0.59)
	SPA-LSSVM	0.98(0.01)	0.32(0.11)	0.92(0.08)	0.75(0.27)	4.00(0.95)

^1^ SPA-PLSR: successive projections algorithm—partial least squares regression; ^2^ SPA-LSSVM: successive projections algorithm—adaptive reweighted sampling.

## Data Availability

The data presented in this study are available on request from the corresponding author.

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
