# Peer review of "Rapid Detection of Different Types of Soil Nitrogen Using Near-Infrared Hyperspectral Imaging"

_molecules, 2022, doi:10.3390/molecules27062017_

Round 1

Reviewer 1 Report

Comments on “Rapid detection of different types of soil nitrogen using Near 2 Infrared Hyperspectral Imaging”

The study has developed a method to use near infrared hyperspectral imaging to quantify the inorganic and organic nitrogen in the soil. The methods seem to be interesting. Some concerns:

  1. in the introduction section, please give some description of machine learning.
  2. what’s the mineralogical composition of the soils used for the experiment? Have you considered the effects of mineral composition when using this method to address nitrogen content in the soil? Need some discussion on the possible effects of mineralogical composition on the precise estimation of nitrogen level based on the spectroscopy methods proposed in the study.
  3. L103-104, the organic matter content may be not different between the two soils. The mineralogical composition may influence the peaks.
  4. Has the authors considered using this method to detect different nitrogen content in natural soils of different nitrogen levels rather than the soils artificially amended with nitrogen?

Reviewer 2 Report

Overall, the paper is well written and contribute to the literature. My specific comments are as follows:

In the abstract, it would be important to explain the main contribution of the paper and indicate the sample and the period.

In the discussion, it would be important to explain the gap in the literature and the contribution of the paper to the literature. It is not clear how the paper contributed to the literature.

In the introduction it would be good to explain the research problem, research hypothesis and contribution of the paper.

The literature review is too sparse. It seems that some earlier work has been cited totally arbitrarily without following a logical plan that could motivate the paper. Please add some of the following references about intelligent algorithms:

Nebojša Denić, Dalibor Petković, Boban Spasić, Global Economy Increasing by Enterprise Resource Planning, Editor(s): Saleem Hashmi, Imtiaz Ahmed Choudhury,. Encyclopedia of Renewable and Sustainable Materials, Elsevier, 2020, Pages 331-337, ISBN 9780128131961, https://doi.org/10.1016/B978-0-12-803581-8.11590-5. (https://www.sciencedirect.com/science/article/pii/B9780128035818115905) 

Boban Spasić, Boris Siljković, Nebojša Denić, Dalibor Petković, Vuk Vujović, Natural Lignite Resources in Kosovo and Metohija and Their Influence on the Environment, Editor(s): Saleem Hashmi, Imtiaz Ahmed Choudhury. Encyclopedia of Renewable and Sustainable Materials, Elsevier, 2020, Pages561-566,ISBN9780128131961,https://doi.org/10.1016/B978-0-12-803581-8.11591-7. (https://www.sciencedirect.com/science/article/pii/B9780128035818115917)   

Denić, Nebojša,  Petković, Dalibor,  Siljković, Boris and  Ivković, Ratko (2020). Opportunities for Digital Marketing in the Viticulture of Kosovo and Metohija. In: Hashmi, Saleem and Choudhury, Imtiaz Ahmed (eds.). Encyclopedia of Renewable and Sustainable Materials, vol. 1, pp. 600–615. Oxford:Elsevier.http://dx.doi.org/10.1016/B978-0-12-803581-8.11592-9     

The research conducted here is not motivated.

Reviewer 3 Report

The introduction has enough information about the rapid detection of different types of soil nitrogen using near-infrared hyperspectral imaging. The methodology is well articulated. The author has written the results with novel information, contribute to the advancement of understanding, and is pragmatic. Discussion of the manuscript is well written and sufficiently discusses the results.

In the overall manuscript, I will suggest that please include specific, detailed comments regarding the originality, scientific quality. Check the needs for tables and figures and the adequacy of the references. Check the spellings. The present work is novel and well written.

Round 2

Reviewer 1 Report

As mineralogical composition in soil may influence the precise characterization of N by NIR, I would strongly suggest the authors take some time to consider and discuss on this aspect if you really want to use this method in the natural soils. The current discussion is not satisfied.

Reviewer 2 Report

accept